# A Systematic Review on Commercially Available Integrated Systems for Forensic DNA Analysis

**DOI:** 10.3390/s23031075

**Published:** 2023-01-17

**Authors:** Brigitte Bruijns, Jaap Knotter, Roald Tiggelaar

**Affiliations:** 1Technologies for Criminal Investigations, Saxion University of Applied Sciences, M.H. Tromplaan 28, 7513 AB Enschede, The Netherlands; 2Politieacademie, Arnhemseweg 348, 7334 AC Apeldoorn, The Netherlands; 3NanoLab Cleanroom, MESA+ Institute, University of Twente, Drienerlolaan 5, 7500 AE Enschede, The Netherlands

**Keywords:** rapid DNA analysis, RapidHIT, ANDE, ParaDNA, forensics

## Abstract

This systematic review describes and discusses three commercially available integrated systems for forensic DNA analysis, i.e., ParaDNA, RapidHIT, and ANDE. A variety of aspects, such as performance, time-to-result, ease-of-use, portability, and costs (per analysis run) of these three (modified) rapid DNA analysis systems, are considered. Despite their advantages and developmental progress, major steps still have to be made before rapid systems can be broadly applied at crime scenes for full DNA profiling. Aspects in particular that need (further) improvement are portability, performance, the possibility to analyze a (wider) variety of (complex) forensic samples, and (cartridge) costs. Moreover, steps forward regarding ease-of-use and time-to-result will benefit the broader use of commercial rapid DNA systems. In fact, it would be a profit if rapid DNA systems could be used for full DNA profile generation as well as indicative analyses that can give direction to forensic investigators which will speed up investigations.

## 1. Introduction

Sampling and securing traces at a crime scene are crucial steps in the investigation process, which in the Netherlands is generally carried out by the police. Information obtained at this stage immediately gives direction to an investigation once the results are available. Nowadays, with commonly applied methods, it might take weeks to obtain from evidence collection to a report provided by a forensic laboratory [1]. Forensic DNA analysis and specifically short tandem repeat (STR) profiling can be improved by performing the first analysis already at the crime scene, e.g., in the situation of disaster victim identification (DVI) [2,3].

The generation of STR profiles currently requires highly skilled geneticists and dedicated laboratories [4]. However, experts from within the chain of criminal law indicate that there is an urgent need for fast and reliable DNA analysis devices for analysis directly at the crime scene. Technical innovations such as the miniaturization of conventional laboratory techniques may both improve their speed and enable their use directly at the crime scene. The latter is extremely important, not only in case of unknown perpetrators or suspect exclusion [2,5,6,7,8] but also when securing traces, as these degrade over time, and returning to a crime scene to collect more evidence is usually not possible.

Forensic DNA analysis, as performed in the Netherlands, can be divided into several sequential operations. Firstly, the evidence must be collected at the crime scene, which is typically done by using a (moistened) cotton swab handled by a forensic investigator. The sample on the swab is sealed in the tube and then sent to the forensic laboratory (usually the Netherlands Forensic Institute (NFI)). The next step is the sample work-up, where cells are eluted and lysed. Subsequently, DNA is extracted from these cells using commercially available kits. Before amplification, a quantification step is usually included within conventional STR profiling since the best results are obtained by the use of a specific amount of input DNA. Quantification ensures maximum efficiency of the amplification reaction and prevents (repetitive) analysis of over-amplified samples. Forensic samples are known for their low amount of DNA, as one cell only contains about 6 pg of DNA. Therefore, it is necessary to perform an amplification reaction to increase the amount of DNA, which ensures detection and further analysis (i.e., STR profiling) can take place [9,10].

Electrophoresis in glass capillaries was already demonstrated in 1981 [11], and the first glass microfluidic chip (also used for capillary electrophoresis (CE)) was published in 1992 [12]. In 1998 Northrup et al. started with the integration of all these steps (sampling, sample work-up, amplification, and detection) within their miniature analytical thermal cycling instrument (MATCI) system, which can be considered the first system towards portability for use at a crime scene. They fitted the complete polymerase chain reaction (PCR) and detection setup into a briefcase due to the integration of the heaters into the systems and the use of silicon reaction chambers (Figure 1) [13]. Within this setup, several amplicons could be successfully amplified and detected in real-time with ethidium bromide.

The MiDAS (short for ‘miniaturized integrated DNA analysis system’) is another integrated system reported in the literature. However, as described by Hopwood et al., further development of this system was discontinued after the closure of the Forensic Science Service in the UK [14,15,16]. The research group of Landers designed, fabricated, and investigated various microfluidic devices for forensic DNA analysis [17,18,19,20], and also Liu et al. reported a fully integrated microfluidic device [21,22,23,24]. Furthermore, Chen et al. made and evaluated an integrated microfluidic cassette for isolation, amplification, and detection of nucleic acids, but not meant for forensic STR profiling [25].

More recently, Zhu et al. developed a droplet PCR combined with multiplex STR for forensics [26]. In their review, Jakaratanopas et al. describe integrated microfluidic systems for genetic analysis [27]. Nouwairi et al. (from the Landers group) developed an injection-molded real-time PCR microfluidic chip. A total of 40 cycles can be completed within 10 min. Note that only one target (TPOX) extracted human DNA from a buccal swab was used. Nouwairi et al. did not integrate sample work-up steps nor multiplexing such as STR profiling [28].

Despite all academic efforts to develop (partially) integrated portable systems for forensic DNA analysis, only a few systems became commercially available for integrated forensic DNA analysis to be used in the field. The ParaDNA, RapidHIT, and ANDE systems are discussed in this systematic review, as well as also some promising devices that never truly became commercially available. A systematic review is carried out to analyze the performance of these systems which are accredited to be used by a forensic investigator on the scene. In the discussion, the technical performance of the analyzed systems is compared, and some critical notes are made about costs and ease-of-use, among other things. Moreover, a critical reflection is given on the use of these systems at the scene, as well as additional societal and organizational challenges.

### Terminology

For clarity and to support readers’ understanding, some words/terms used in this work are explained here. 

An allelic drop-out is a failure to determine an allele within a sample. This occurs when one or both allelic copies are not amplified during PCR.PCR inhibitors are chemical or physical obstacles that block amplification and, as such, ultimately fail the PCR reaction.‘Rapid DNA analysis’ is defined as using a rapid DNA instrument without human intervention, whereas ‘modified rapid DNA analysis’ is defined as using a rapid DNA instrument in combination with human interpretation of the DNA analyses results [2,29,30,31].Rapid DNA systems are fully integrated platforms that can generate STR profiles from (reference) samples within two hours [32].

## 2. Literature Search

### 2.1. Search Strategy

For the systematic review, a search for articles was carried out in the PubMed database without language or date restrictions. The following keywords were used: (ParaDNA) OR (RapidHIT) OR ((ANDE) AND (DNA)). The articles that were included ‘ad hoc’ were found via Google Scholar or used as a reference in one of the articles from the systematic review.

A ‘Preferred Reporting Items for Systematic Reviews and Meta-Analyses’ (PRISMA) flow diagram is used to summarize the screening process of the scientific literature [33]. It makes the selection process transparent by reporting on decisions made at various stages of the systematic review. For inclusion in this review, studies are excluded if they are not about STR profiling, not about human DNA, nor (yet) peer-reviewed.

### 2.2. Search Results

The results of the literature search, as described in Section 2.1, are depicted in Figure 2. The literature search in PubMed resulted in 16, 20, and 24 hits for the ParaDNA, RapidHIT, and ANDE systems, respectively. In the case of ParaDNA, two articles were added through ad hoc search, and for RapidHIT, this search resulted in five additional publications.

Subsequently, the articles were screened by title and abstract. For ParaDNA, seven articles were excluded since they did not involve STR analysis/DNA profiling but described ParaDNA Body Fluid ID Test (see for more information Section 3.1). Another three articles were excluded because they dealt with an unrelated topic using the same HyBeacon technology as the ParaDNA system. Upon screening of the hits for RapidHIT, one article was excluded since it was in Chinese and not available in either English or Dutch. For the ANDE systems, three articles were excluded, as they described an unrelated topic. Another six hits were excluded since they appeared because their topic was DNA related or because one of the authors was called ‘Ande.’

In the next phase (the eligibility phase), the articles were fully read. In total, four studies were excluded: one manuscript was not peer-reviewed (yet), one article described an old system (the predecessor of the ANDE system, further discussed in Section 3.4), and two articles overlapped (they described both the RapidHIT and ANDE).

Post to the eligibility phase, the number of articles selected for inclusion in this review are 8, 21, 12, and 2 for ‘ParaDNA,’ ‘RapidHIT,’ ‘ANDE,’ and ‘RapidHIT & ANDE,’ respectively.

## 3. Rapid DNA Systems

The use of DNA analysis is of enormous forensic interest since every human being has unique DNA. Nowadays, DNA from, e.g., fingerprints, a bit of saliva, or a small bloodstain can be collected and analyzed [34].

Conventional DNA analysis is costly and time-consuming as DNA extraction takes about 30 min, STR-PCR about two hours, and CE another hour of run time. It must be noted that the mentioned times are only an indication (and are the minimum required for each step) because the exact time depends on, e.g., the type of sample (teeth and bone require longer extraction times) and laboratory protocols. When DNA quantification is included, another two hours of thermal cycling is needed, thus leading to a time-to-result in the range of 3.5 to 5.5 h or even more [30,35]. A fully integrated system to perform forensic DNA analysis at the crime scene can save both time and costs [34,36]. Such a system should, among others: (a) be easy to use, also by minimally trained people, (b) make use of disposable cartridges for single use, (c) have a fast time-to-result (preferably under 1 h), (d) be portable, (e) be an enclosed system (i.e., no operator manipulation possible as to avoid contamination), (f) be robust (i.e., transportable), (g) be able to analyze a variety of forensic samples, (h) perform well (sensitivity and selectivity), and (i) not be too costly [35,36].

In January 2013, an article was published in ‘Biometric Technology Today’ about the use of rapid DNA technology, i.e., full DNA analysis within 2 h. At that time, two systems were available for commercial use: RapidHIT 200 and ANDE 4C. The FBI and the US Army started trials and evaluations [37]. 

In the following sections, the rapid DNA systems—ParaDNA, RapidHIT 200, its successor RapidHIT ID, and the ANDE system—that made it to market are reviewed.

### 3.1. ParaDNA

The ParaDNA Intelligence Test System has been developed by LGC (Laboratory of the Government Chemist, Teddington, UK). ParaDNA screens five STR loci (D3S1358, D16S539, D8S1179, D18S51, and TH01) and the sex marker Amelogenin [38]. The time-to-result of the ParaDNA system is 75 min, and this system is based on HyBeacons technology and melts curve analysis [39]. These beacons are made of short DNA sequences with one or more fluorescent dyes, which will emit light when the complementary DNA is attached to the beacon. The system consists of a screening unit, a sample collector, and a test kit [38,40]. The collector is used similarly to a swab; its head consists of four plastic tips (used for sampling of (potential) traces) (Figure 3). Post to sample collection, the head of the collector is inserted into a 4-well reaction plate [40].

The ParaDNA system is a so-called ‘Level 1 DNA Screening Solution’, which implies that it meets the following criteria: (1) the test must be a DNA test, (2) it should provide the police with a more robust and reliable decision making tool, (3) should speed up early intelligence, (4) is easy-to-use by non-scientific personnel, and (5) the test result must be interpretable by minimally trained personnel [40].

Li et al. validated the ParaDNA test with mock evidence samples, according to the Scientific Working Group on DNA Analysis Methods (SWGDAM) guidelines. They used blood on glass, on Flinders Technology Association (FTA) cards and on denim, saliva on glass, buccal swabs, drink bottles, smoked cigarettes, semen on glass and carpet, and touch DNA on clothing and on a mobile phone. Both the blood and saliva samples had a success rate of 100%. When using saliva on swabs, the success rate dropped to 80%. Semen samples and touch DNA had a success rate of 60–75% (depending on the analysis software) and 14%, respectively [38].

Tribble et al. tested the ParaDNA Screening System with seminal samples, which profiles D16S539, TH01, and Amelogenin loci. They tested serial dilutions of semen, and at 2 nL of semen in each well, the assay showed a drop in Amelogenin. At 0.2 nL, the presence of DNA was no longer detected [39]. Ball et al. tested 381 DNA samples, and only 0.15% was discordant. Dropout only occurred in samples with a heterozygote imbalance or in cases of high stutter [42]. Donachie et al. concluded that the ParaDNA system could be used as a presumptive test for samples with medium to high template DNA present, such as blood, saliva, and semen stains. For low template samples, such as touch DNA (e.g., single fingerprints), the success rate was too low [43].

Dawnay et al. report results with the ParaDNA screening system as obtained during the UK Police pilots’ phases III and IV (in which validation of the future operating procedure of this system in police hands by various Police Forces took place). A profile was defined as successful when 14 or more alleles were determined. For blood, the success rate was 100% in both phases. When using saliva as a sample the success rates were 88.9% and 82% for phases III and IV, respectively. Touch samples scored relatively low, viz. 33.3% in phase III and 65.2% in phase IV [40]. They determined that upon using this system for touch DNA samples, 28/34 samples would result in a successful STR profile, while without this technique it would be 42/83. Therefore they state that the ParaDNA system will increase profiling success rates, reduce backlogs and result in cost savings while noting that it is only a presumptive test and cannot replace existing STR profiling [44]. Dawnay et al. also performed field tests to determine the impact of sample degradation and inhibition when using human remains as a sample. They valued the possibility of performing rapid processing in the field with the ParaDNA system. However, further optimization is required concerning the collection process in the case of human remains [45]. 

Using 4 ng of more DNA (corresponding to 1 ng per well), Blackman et al. showed that more than 65% of the samples provided a full profile. A ‘usable’ (≥ 7 alleles detected) profile could be generated with 250 pg of DNA in more than 80% of the samples by using the ParaDNA system [46].

In conclusion, the ParaDNA Screening System can be used for samples that contain a relatively high amount of DNA, such as blood, saliva, and semen [38,39,40,43]. Since only 6 STRs (or less for some kits) are analyzed, the ParaDNA system can only be used for screening and indicative testing, which is a limitation of this system [45]. 

Later on, the so-called ParaDNA Body Fluid ID Test became available. This test also uses of HyBeacon probes and is an RNA-based test to determine the type of body fluid (seminal fluid, sperm cells, blood, saliva, vaginal fluid, and menstrual blood) [41,47,48,49]. In 2013 ParaDNA was acquired by Life Technologies Corporation [50]. However, the production of ParaDNA is currently discontinued [51].

### 3.2. RapidHIT

The RapidHIT ID (launched in 2013), a system from IntegenX (which is part of Thermo Fisher Scientific as of 2018), is a cartridge-based system and can create a full STR profile within 90 min (Figure 4). The GlobalFiler analysis results are acquired by integrating DNA extraction, PCR amplification, and CE in one machine [4,6,52,53,54,55,56,57,58]. Prior to the RapidHIT system, IntegenX developed the Apollo 200 DNA HID System, which integrated cell lysis, DNA extraction, amplification, separation, and detection from buccal swabs or blood samples [59]. Another predecessor was the RapidHIT 200, which was later on re-branded to the RapidHIT Human Identification System, whereby the lysis temperature was somewhat adapted [2,56]. The RapidHIT 200 contains the same chemistry as the RapidHIT ID and can generate STR profiles within 2 h [29]. Thermo Fisher only sells the RapidHIT ID nowadays, which is an FBI NDIS-approved rapid DNA system [60]. 

The ACE cartridge is used for the analysis of a single swab (e.g., a reference buccal swab), and the EXT cartridge is for the analysis of DNA extracts (evidence profile) [61]. Later on, there were two types of cartridges available: the ID ACE GlobalFiler and the RapidINTEL for oral cells and minute samples (e.g., blood stains), respectively [52,57]. Although RapidINTEL was originally developed for the analysis of blood and saliva samples, it has also been validated for other tissue samples [52,55,62]. Nowadays, the RapidHIT ID uses two chemistries/kit types: the GlobalFiler Express cartridge and the AmpFLSTR NGM SElect Express cartridge. GlobalFiler chemistry for buccal swabs (the ACE GlobalFiler Express Sample Cartridge) and for blood and saliva samples (the RapidINTEL GlobalFiler Express Sample Cartridge) can be ordered on the website of ThermoFisher. The AmpFLSTR chemistry is available as ACE NGM SElect Express Sample Cartridge to be used for buccal swabs. The EXT cartridges are not available under that name anymore [63].

RapidHIT systems are evaluated extensively in the literature. Articles that describe which system is used are discussed in the corresponding Section 3.2.1 and Section 3.2.2, while articles in which the used system type(s) are not clearly mentioned are evaluated below.

A sensitivity level of 176 ng saliva DNA was reported by Jovanovich et al. for full concordant profiles and using 16 ng of DNA (on a swab), 57% of the alleles could still be detected. A 100% success rate was achieved for the positive controls (37 samples). Out of 250 buccal samples, 219 samples were fully in concordance with the PowerPlex 16 profile on the first pass [64].

Hennessy et al. assessed the sensitivity, accuracy, and genotype concordance of this RapidHIT system. Full profiles were obtained till 500 pg when DNA was added to the vials; down to 25 pg, 65% of the alleles were detected. In case DNA was added to swabs, a sensitivity of 100 ng was found. Moreover, when 5 ng DNA was loaded into the swabs, 52% of the alleles were detected. It has to be noted that all standards tested were in 100% concordance with the certified reference profiles. For 150 buccal swab samples, a 100% genotype concordance was obtained; the 13 Combined DNA Index System (CODIS) core loci were present on 94.7% of the samples [62].

Buscaino et al. could obtain complete profiles with 75 pg/µL for GlobalFiler Express and 50 pg/µL for AmpFlSTR NGM SElect Express and a total amount of input DNA (depending on the sample input volume) less than 250 pg. Five samples were run in triplicate, producing complete and concordant STR profiles, thus showing good reproducibility. The AmpFlSTR NGM SElect kit showed full concordance for all tested mock crime scene samples, while the GlobalFiler Express kit, 5/7 samples did not result in full concordance [65].

Martin et al. investigated the use of rapid DNA analysis on touch DNA samples. Samples such as cable ties, fabric, matchstick, and ziplock bags were analyzed. The cable ties and matchsticks were directly placed into the cartridge of the RapidHIT. Together with tape-lifted fabric, these samples showed the best results [66].

The RapidHIT was extensively evaluated by LaRue et al. Among other things, they tested the variability, sensitivity, concordance, and reliability, and they performed a contamination study. The success rate using reference buccal swabs was comparable to standard methods. Full profiles were obtained with an input of 200 ng of DNA, while with 10 ng, more than 10% allelic drop-out was observed, resulting in partial profiles [67].

#### 3.2.1. RapidHIT 200

Thong et al. compared the performance of the RapidHIT 200 System (in combination with the GlobalFiler Xpress Cartridge) with the standard protocol. They tested swabs with blood, semen, and buccal samples, blood-stained FTA punches, tissues, bone marrow, fingernails, and cigarette butts. Overall, the RapidHIT 200 System performed well compared to standard laboratory protocols. It was recommended by Thong et al. to perform more experiments with more challenging DNA samples, such as degraded samples, samples containing inhibitors, mixtures, and samples with a small amount of DNA [68].

Shackleton et al. investigated the performance of the RapidHIT200 with crime stain type samples on the RapidHIT 200, which contain various (and compared to buccal swab samples lower) input amounts of DNA. Extending the sample work-up time (DNA bead binding time or lysis time) had a positive effect on the results, but this negatively affected the time-to-result: i.e., up to 30–60 min extra time was needed [69]. Shackleton et al. validated the RapidHIT 200 further for accreditation and used it in the UK. Sensitivity down to 4.5 ng (750 cells) on a swab was obtained, while for lower cell loads, allelic drop-out was observed. They further demonstrated that the AmpFlSTR NGMSeE kit, in combination with the RapidHIT 200, is a reliable method for obtaining reference profiles from a buccal swab [70]. 

The RapidHIT 200/GlobalFiler Express system (with RH200/GFE cartridge kits) was evaluated by Date-Chong et al. The built-in quality flags in the software appeared, among others, in cases of intra-locus imbalance and inconclusive homozygote allele. Of all reference buccal samples, 50% passed the criteria of generating a full profile without raising any quality flag, which would require manual editing. With manual data review, the success rate is increased to 91% for obtaining a full genotype at each locus [71].

Holland et al. also evaluated RapidHIT 200 for automated human identification of single source samples. A total of 85 buccal swabs were analyzed, and 100% concordance was obtained with known DNA profile information. When using the software of the instrument for data analysis, 95.8% provided full profiles in the reproducibility study [56].

The RapidHIT 200 was explored by Verheij et al. with different types of samples. They concluded that the system is very suitable for buccal swabs. Moreover, saliva, semen, skin, and hair samples were tested, but profiling success rates were variable, and with lower input samples, profiling artifacts were present. The blood samples tested presumably contained inhibitors since no profile could be obtained [72].

Only 74% of the buccal swab samples that were tested by Mogensen et al. on a RapidHIT 200 resulted in a correct STR profile. The working ranges in order to generate full profiles were found to be 900–1200 ng. Lower and higher amounts of DNA resulted in allelic drop-out [73].

#### 3.2.2. RapidHit ID

Shackleton et al. used the AmpFlSTR1 NGMSElect Express STR kit with single source reference samples. Prior to testing the performance of the RapidHIT ID, the instrument protocol was optimized by adjusting, among others, the lysis temperature, the thermal cycling parameters, and the injection time for CE. The repeatability, reproducibility, and sensitivity were found to be sufficient for buccal swabs (obtaining a reference profile) [54].

Amick et al. validated the ACE and evaluated the EXT cartridges according to the Quality Assurance Standards from the FBI. Among other things, they investigated the reproducibility, precisions, and sensitivity and performed a contamination assessment. The RapidHIT ID, using ACE cartridges, showed reproducibility and precision within a ±0.5 bp window. According to Amick et al., the ACE cartridges showed acceptable results for buccal swabs. The sensitivity test of the EXT cartridge revealed allelic drop-out for input of 2500 pg DNA and lower. The main drawback of the EXT cartridge is that an evidence sample must first be extracted and quantified before analysis on the RapidHIT [61].

The use of blood samples in combination with the RapidINTEL cartridge and the RapidHIT ID System was investigated by Chen et al. They also concluded that FTA cards are the preferred method for storage. Urea and melanin cause inhibition resulting in signal decrease and allelic drop-outs. Using 0.5 µL of blood or 7 ng of genomic DNA resulted in a concordance rate of 100% [53].

Murakami et al. evaluated the performance of the RapidHIT ID (with ACE and INTEL cartridges) with samples from postmortem bodies at different stages of decomposition. Moreover, nail clipping samples were collected and analyzed. A detection rate of 100% was obtained with the latter, even after 10 years of storage, as long as the nail clippings were pretreated with dithiothreitol and TNE (Tris, NaCl, and EDTA) buffer. For blood stains, the recommended storage method is using filter paper at room temperature [52].

A success rate of 72% was reported by Wiley et al. upon testing the RapidHIT ID (cartridge with GlobalFiler Express primer set and master mix). After manual interpretation of first-pass inconclusive results, the success rate could be increased up to 90%. While coffee, smoking tobacco, and chewing tobacco did not influence the result, the swab type used is of importance since it can affect the typing success rate [4].

The RapidHIT system was used by Gangano et al. to demonstrate the usability of the system with blood and saliva samples. They concluded that 88% of the samples (blood and saliva) generated DNA profiles that were classified as qualitatively sufficient enough to be able to search against a database. Moreover, a reanalysis study was performed where the swabs were recovered from the cartridge after analysis on the RapidHIT. Reanalysis of these samples on the RapidHIT demonstrated that conserving these samples is possible with an average of 92% of the original full loci calls [74].

Upon use of the GlobalFiler Express kit, Salceda et al. found a sensitivity of 12,500–200,000 cells/swab, while for 6250 cells, only 51.6% of the alleles could be detected. Five control samples were tested with three RapidHIT ID machines, and all 15 profiles were complete and concordant. The authors claim that this system showed 100% concordance in comparison with conventional methods [75].

Ward et al. used the RapidHIT ID system and RapidINTEL sample cartridges to evaluate the performance with mixtures and touch DNA samples. The mixtures were artificially constructed from raw DNA as well as from body fluids. While the profiles from the simulated DNA mixtures on the RapidHIT showed the expected mixture proportions, the peak heights were lower compared to results when using a standard laboratory workflow. For the mock casework, mixtures, blood, saliva, or both blood and saliva were used. A combination of saliva and blood led to low levels in the profile of the saliva contributor. The sensitivity of the RapidHIT is lower than a standard laboratory workflow, which affects the results when touch samples are analyzed. Many of the analyzed touch DNA samples produced very low-level DNA profiles [76].

Cihlar et al. validated the RapidHIT ID system with the ACE GlobalFiler Express sample cartridges in combination with reference buccal swabs. Among others, they performed a concordance, contamination, sensitivity, reprocessing, and mixture study. The data generated showed that the workflow of RapidHIT generates reliable and reproducible DNA profiles. A first-pass success rate of 92% was obtained with the reference buccal swabs. However, Cihlar et al. also advise using a trained forensic analyst to increase the pass rate success [57].

In conclusion, the RapidHIT 200 provides good results in the case of single-source reference samples, especially when manual evaluation is performed [56,71]. A limitation of the system is the inability to quantify the amount of DNA that is added to the PCR step [56]. The RapidHIT 200 was later on replaced with the RapidHIT ID, which also shows good results for reference buccal samples when combined with the ACE cartridge. Wiley et al. and Cihlar et al. recommend performing a manual evaluation of the generated DNA profiles, especially for case samples [4,57].

### 3.3. ANDE

Before the ANDE systems became commercially available in their current format, it was developed and tested by NetBio, which was published as a research report in 2009 [36]. In 2013 Tan et al. reported on a fully integrated system for the automated generation of DNA profiles. With the so-called BioChipSet cassette, a total of 5 buccal swabs could be analyzed simultaneously. The system integrated lysis, purification, amplification, CE separation, and detection. The generated profile involves 16 loci and is generated in 84 min. The PCR mixture is present in the cartridge in lyophilized format. A full CODIS profile was obtained with 85 of the 100 buccal samples analyzed. The other samples resulted in a partial profile (5) or no profile (10). Blocked channels were the cause of failure, resulting in no amplification or failing electrophoresis [77]. In order to be able to generate profiles from lower amounts of DNA, a Low DNA Content BioChipSet cartridge was developed [78]. Note that the system itself was already known as ANDE [77,78]. In order to generate profiles from a minor amount of DNA, the chemistry was adapted, and a purification module was integrated into the cartridge, while the system (both analysis and read-out) remained identical to a great extent. Full profiles could be generated with 1 µL or more blood as an input sample in the sensitivity study [78]. A study was carried out with eight laboratories (including NetBio), which tested over 2300 swabs in total with the DNAscan/ANDE Rapid DNA Analysis System to be compliant with the FBI’s Quality Assurance Standards and the NDIS Operational Procedures. Concordant results were obtained for the reference buccal swabs, including automated data analysis and an accuracy allele calling rate of 99.998% [79].

In 2012, DNAScan was the first system released by ANDE (‘Accelerated Nuclear DNA Equipment’), which was later renamed to ANDE 4C (4-color). However, the ANDE system was originally developed by NetBio [58]. The system integrates the steps of DNA extraction, STR amplification, separation by electrophoresis, and detection and performs them all within 90 min [80]. The A-Chip for the ANDE 6C (6-color), which can process up to five buccal samples, was previously known as the BioChipSet cassette [2,58,80,81]. The so-called I-chip, depicted in Figure 5, is specifically designed for low-template DNA samples and can process up to four samples simultaneously [81]. The ANDE 6C system uses a Bode SecureSwab2 with an integrated RFID chip for sample tracking in the swab cap and is able to analyze 27 loci [80]. The ANDE 6C received NDIS approval in 2018 [30].

In addition to DVI, trafficking in persons is also a situation in which rapid DNA results are crucial. Palmbach et al. investigated whether the ANDE system (at that time still part of NetBio) was suitable for this purpose. A success rate of 95% was obtained for the reference profiles; the other 5% resulted in partial profiles. The other samples consisted of various objects, such as cigarette butts, condoms, plastic bottles, and straws. Overall, a success rate of 71% was obtained, consisting of 23%, 42%, and 6% for full profiles, partial profiles, and mixed profiles, respectively [82].

Moreno et al. evaluated the first version, the DNAscan, which generated a profile from CODIS 13 loci. Without human review, an overall success rate of 75% was achieved from reference buccal swabs [32].

The developmental validation of the DNAscan/ANDE system was carried out by Della Manna et al. using a BioChipSet Cassette. Over 2000 buccal swabs were analyzed, which resulted in over 99.998% concordant alleles [79].

The California wildfires in 2018 were the motivation for Gin et al. to use rapid DNA for victim identification. Among others, blood, as well as liver, brain, and muscle tissue, were used for DNA profiling. In 90% of the cases, the ANDE (no type mentioned) led to the identification and was even the primary identification modality [83].

#### 3.3.1. ANDE 4C

The ANDE (probably 4C) system was used by Hinton et al. for the analysis of reference samples (buccal swabs), samples from a controlled setting, and from an uncontrolled environment. In combination with the A-Chip, 19 of the 22 buccal swabs resulted in a complete profile. For the controlled setting, the I-Chip was used, which provided a full profile for 7 of the 12 samples. The I-Chip was also used within military exercise, i.e., the uncontrolled environment. Only 7 out of 44 samples could generate a full profile from a single contributor. The other samples resulted in no DNA profile (25), partial DNA profile (9), and DNA mixtures (3) [84].

#### 3.3.2. ANDE 6C

Ragazzo et al. validated the ANDE 6C according to the ISO/IEC 17,025 standard. A total of 104 buccal swabs were analyzed with the ANDE system and with the traditional laboratory method. Of these 104 samples, three swabs gave no interpretable signals. A concordance rate of 99.96% was obtained by comparison of 2800 genotypes. With these results, the aforementioned standard was met [80]. 

While the ANDE 6C was originally designed for reference samples (i.e., buccal swabs), Manzella et al. tested the performance of this system with calcified and soft tissue samples. The success rate was 0%, 11%, and 50% for muscle tissue, ribs, and teeth, respectively [85]. 

Turingan et al. analyzed 1705 casework samples to analyze the performance of the ANDE 6C with the I-Chip to address the FBI’s Quality Assurance Standards. Among other things, they investigated specificity, sensitivity (including a limit of detection), reproducibility, mixtures, and a wide variety of sample types (e.g., dried blood, blood on various substrates, semen (neat and stains), and saliva). Their main conclusion was that the limit of detection is influenced by the type of sample, the condition (aging and degradation), and the substrate [81]. 

Six forensic and research laboratories worldwide took part in the validation study of the ANDE 6C of Carney et al. Over 99.99% concordant alleles were generated with the over 2000 analyzed samples (buccal swabs in combination with the A-Chip), without human review of the results [86].

Also, Grover et al. evaluated the ANDE 6C system and the A-Chip, using only buccal swab samples. Among others, they tested sensitivity, reproducibility, species specificity, and stability. Overall, they concluded that the system performs well and that the automated analysis results in accurate and concordant DNA profiles [31].

Manzella et al. used the ANDE 6C system to investigate the difference between the ANDE swab and a conventional cotton swab as a sampling method. Using both swabs for buccal samples resulted in a success rate of 33% and 88% for the conventional cotton swab and the ANDE swab, respectively [85].

Turringan et al. used the ANDE 6C to identify human remains. They used several tissue types, such as the buccal, brain, liver, and muscle. Up to 11 days of exposure, buccal swabs are the sample of choice. Moreover, bone and tooth samples provided good results for up to 1 year (which was the duration of the study). When the remains are refrigerated, all tissue types yield good results [87].

In conclusion, the ANDE systems (4C and 6C) work very well when buccal swabs are used as input, even without human intervention (i.e., rapid DNA analysis) [86]. While being able to have rapid DNA profiles in a situation such as DVI is extremely useful, these types of samples show a lower success rate compared to reference samples [85].

### 3.4. Other Systems

NEC developed a portable DNA analyzer that can perform DNA analysis within 25 min, with 5, 15, and 5 min for DNA extraction, PCR, and CE, respectively. The disposable cartridge contains 5 mm wells that are used as test tubes. The channels for fluid transfer serve as pipettes. The system weighs 32 kg and can analyze 16 loci. It must be noted, however, that the most recent publication using the NEC system dates from 2012 [88].

As already mentioned in the introduction, MiDAS is a microfluidic system for rapid forensic DNA analysis. In contrast to other rapid DNA analysis systems with all reagents present in the cartridge, MiDAS requires manual preloading of all reagents onto the cartridge. Moreover, other manual steps are required in the operation of the system, such as the installation of the CE chip. The system consists of a total of five interdependent modules: (1) swab lysis, (2) DNA extraction, (3) PCR, (4) PCR product transfer, and (5) CE separation, including optical readout [89]. Each module is discussed separately in various publications. The lysate was initially used as input, but after several improvements, the swab head could be used in combination with the swab sample lysis module [15]. After insertion of the swab in the module, lysis occurs. Then, the lysate is transported to a chamber for purification, amplification, and detection [90]. Hopwood et al. described a cartridge that integrates DNA extraction (from lysate), amplification, CE, and detection using laser-induced fluorescence (Figure 6). With this system, it is possible to obtain an STR profile within 4 h [14]. The PCR module is discussed by Estes et al. for the amplification of 17 STRs. This cartridge uses preloaded solid-phase reagents and paraffin valves for fluidic control [91]. The electronic control components, the CE microchip, and the optical detection module are described by Hurth et al. [16].

## 4. Discussion

In this section, various aspects of commercial systems are discussed, among others, their performance, portability, and costs. For comparison (and overview) purposes, Table 1 is composed. Since quite a lot of the literature is available regarding the RapidHIT 200, this system is included in the table, whereas this is not the case for the ANDE 4C system.

Besides technical challenges, there are quite some societal and organizational hurdles to overcome upon using (modified) rapid DNA directly at the crime scene. Jurisdiction and legislation are important parameters that influence the successful use as well as utilization of rapid DNA techniques. In the Netherlands, the forensic investigator arrives at the crime scene to investigate what has happened, how it happened, and especially who did it. To find out the latter and to match a DNA profile to a suspect, the taken samples are normally sent to a forensic laboratory such as the NFI. This process takes up much precious time. The forensic investigator is not aware of whether the collected samples contain (enough and non-degraded) DNA and if the sample is from the perpetrator(s) or victim(s). In the meantime, a tactical investigator would like to have evidence to connect a suspect to a crime scene. In order to obtain a more efficient coupling between forensic and tactical investigators, fast results are essential, which could be accomplished by means of using (modified) rapid DNA technology for generating full DNA profiles as well as for screening samples.

### 4.1. Performance

Rapid DNA techniques show good performance when single source reference samples (buccal swabs) are used. Moreover, in the case of blood and saliva (samples with a relatively high DNA content), a full DNA profile can be expected. However, it is known that rapid DNA techniques are less sensitive than conventional laboratory DNA analysis processes. This is caused by the conventional process having more sample work-up steps (i.e., extraction and/or purification) and using more sensitive machines [5,66]. Additionally, rapid DNA techniques are not or less suitable for resolving DNA mixtures [5]. This results in a trade-off between performance on one hand and speed and portability on the other [76]. 

### 4.2. Direct Comparison

A few publications compared the RapidHIT and the ANDE systems. The RapidHIT ID System and the ANDE 6C Rapid DNA System were compared by Watherston et al. for DVI simulations [55]. Moreover, Romsos et al. tested the ANDE and the RapidHIT (both the RapidHIT 200 and the RapidHIT ID) with sets of blinded single-source reference samples [29].

Nail and quadriceps tissue turned out to be the best sample types for DNA profiling in the case of DVI. Nail samples were also found to be an appropriate type of sample. Overall, both systems were suitable for identification at the (crime) scene. While the ANDE 6C system is more robust in terms of resistance to shocks upon transport (to a crime scene), the RapidHIT can generate more DNA profiles that are usable [55].

Romsos et al. concluded that modified rapid DNA analysis is needed for both RapidHIT systems [2,29]. The rapid DNA analysis method resulted in a success rate of 80% for full profiles. For the modified method, a success rate of 90% was obtained [29].

### 4.3. Portability, Time-to-Result, and Throughput

The National Police in the Netherlands has purchased a RapidHIT for research purposes, as described by de Roo et al. This RapidHIT is placed in a dedicated advanced bus to facilitate an environment that matches the applicable quality requirements for DNA research [5]. Although the RapidHIT ID system has the lowest weight (29 kg), two people are needed for transportation. Locating such a system in a bus or van makes it mobile but not portable.

Another important parameter for being a truly rapid method is the time-to-result (in combination with throughput). The RapidHIT ID is the fastest system, with a time-to-result of 90 min, although the ANDE 6C, in combination with the A-Chip, is not much slower, with 94 min. While this is truly a record in terms of time needed for generating a profile from a swab as input, the limited throughput of 1 and 5 samples per run for the RapidHIT ID and the ANDE 6C, respectively, is a bottleneck for analysis at a (complex) crime scene such as DVI situations.

### 4.4. Sensitivity

The sensitivity (minimum input required, determined with controlled cell or DNA dilution series, whereby full profiles consistently were generated) varies between the different systems (Table 1). The replacement of the RapidHIT 200 with the RapidHIT ID resulted in a gain in sensitivity range from 50–200 ng to 40–80 ng. For the ANDE systems (4C and 6C), a broad sensitivity range of 250 ng–2 µg has been reported [2]. With conventional laboratory techniques, as little as 100 pg is enough to generate a DNA profile [9].

### 4.5. Costs

According to Watherston et al., a sample run on the ANDE and RapidHIT costs 387 and 247 Australian Dollars (~€250 and ~€169), respectively [55]. The RapidHIT ID Primary Cartridge NGM Select Kit costs €7700 for 150 kits (€51.34/kit); 10 cartridges of the RapidINTEL Sample Cartridge Evaluation Kit are €2000 (€200/kit). The RapidINTEL Sample Cartridge Kit is available for €6010 per 50 cartridges (€120.20/kit). The price of the RapidHIT ID system is not publicly available on the website of Thermo Fischer Scientific but can only be obtained via a quotation request [92]. Since the ParaDNA system is not commercially available anymore, no (actual) price indication can be given [51]. Morgan et al. investigated the potential investigative value of a decentralized rapid DNA workflow for reference samples. They concluded that the costs of rapid DNA technology are significantly higher than the current laboratory workflow. Only when costs are reduced they predict that rapid DNA technology can possibly become a viable option in Australia [6].

### 4.6. Practical Challenges

Although the use of commercially available systems for DNA analysis is growing, there are (still) various practical challenges to overcome.

In general, non-expert users are able to run a rapid DNA system (e.g., process samples, identify flagged issues and re-run samples if needed). However, for more complex samples, due to, e.g., inhibition or degradation, expert review is required. Additionally, rapid DNA analysis is less sensitive and shows more allelic drop-out compared to conventional DNA analysis due to the limited sample work-up, resulting in limited removal of dirt or soil and the presence of inhibitors, among others [5,55].

Watherston et al. mentioned that the RapidINTEL cartridges for the RapidHIT required refrigeration (i.e., storing −4 °C until use) near the testing area, while the I-Chip for the ANDE could be stored at room temperature [55]. 

As can be seen in Table 1, only a limited amount of samples can be analyzed per run and, therefore, also per day. While, in theory, rapid DNA analysis is highly suitable in DVI cases, the limited throughput is a hurdle.

### 4.7. Jurisdiction and Legislation

Using rapid DNA systems at a crime scene involves some jurisdictive hurdles. According to Dutch law, DNA analysis can only be conducted by an accredited laboratory, and only a forensic expert is allowed to perform the interpretation of DNA profiles [10]. In order to make it possible for the Police to perform DNA analysis with a rapid DNA system, the Dutch law was adopted on 1 November 2020 [93,94]. However, according to de Roo et al., currently, the rapid DNA system cannot be used without additional quality control. Therefore, de Roo et al. used the raw data from the RapidHIT to generate DNA profiles at the NFI [5]. 

In the USA, the law was adopted somewhat earlier since the Rapid DNA Act was already signed in 2017. The aim was to reduce the DNA analysis backlog in the USA and to reduce violent crime. Because of this adaptation, it is allowed to perform DNA profiling with rapid DNA systems of arrestees in booking station environments [35]. 

Wilson-Wilde et al. mention the legislative hurdle that plays in Australia. Within the law enforcement context, there are only three locations at which rapid DNA systems can use in Australia. Besides, within the existing forensic laboratories (which can analyze all sample types), it is also possible to analyze crime scene samples within Police crime scene units (with a possible backlink to a forensic laboratory) and within police stations. In the latter case, only reference samples are allowed [3].

### 4.8. Indicative Testing

The ParaDNA system cannot generate a complete DNA profile, only 6 STRs [38]. This makes the system an indicative test. Investigators at the crime scene, both forensic and tactical, benefit from results that contribute to guiding information. This result does not necessarily have to be a full DNA profile as long as a scenario can be confirmed or refuted. It would be enormously helpful if an investigator knew if the victim(s) or the perpetrator(s) were the source of the evidence. First screening at the crime scene can result in earlier identification of a possible suspect. In this case, the suspect has less time to commit another crime, destroy evidence, put an alibi on the scene, or come up with an alternative scenario. For a successful implementation, speed and costs are key. Such a screening system can contribute to reliable decision-making and should speed up early intelligence [40]. 

## 5. Conclusions

The use of (modified) rapid DNA systems on the crime scene involves a trade-off between speed and performance. In general, rapid DNA systems are not as sensitive as conventional laboratory methods. While the speed and portability are extremely useful for, e.g., DVI cases, their performance (in terms of sensitivity and the ability to use samples other than buccal swabs) has to be (further) improved to actually use these techniques on crime scenes. The discussed systems work well when using single-source buccal swabs (i.e., reference samples), but for more complex samples, the success rate (i.e., generating a full DNA profile) is too low. Below, conclusions regarding currently available commercial systems for rapid DNA analysis are provided by focusing on multiple aspects/requirements (as listed in Section 3).

Ease-of-use: although the rapid DNA systems themselves can be operated by minimally trained people, the interpretation of the DNA profiles gives a higher success rate when this is done by an expert (i.e., modified rapid DNA), especially for the RapidHIT.Use of disposable cartridges for single use: All (modified) rapid DNA systems make use of sample cartridges that are single-use cartridges.Time-to-result: although it is not (yet) possible to generate a DNA profile within the targeted 60 min, obtaining a DNA profile in ~90 min is possible.Portability: With a weight of at least 29 kg and the present dimensions, current rapid DNA systems are not really ‘briefcase’ sized systems. When loaded into a bus or van, such systems become mobile but not portable.An enclosed system (no operator manipulation possible to avoid contamination): All rapid DNA systems fulfill this requirement.Robustness: The rapid DNA systems are robust in terms of being transportable. They can withstand, e.g., shocks while being transported from one location to another in a car or van.Possibility to analyze a variety of forensic samples: The RapidHIT works well with buccal swab samples, but for other types of samples, the success rate drops. While the ANDE performs somewhat better with real case samples, reference samples (i.e., buccal swabs) are the preferred sample type.Performance (sensitivity and selectivity): Even though the selectivity of the rapid DNA systems is not widely reported, no incidents were mentioned. Their sensitivity, on the other hand, is still substantially lower than conventional laboratory methods. It has to be mentioned that a trade-off exists between sensitivity and speed: in general, a higher analysis speed negatively influences the sensitivity that can be obtained with a system.Costs: About €200 per cartridge run makes the use of rapid DNA systems quite costly, especially with the low throughput/amount of sample per cartridge taken into account.

Both systems, the ANDE and the RapidHIT (ID) suffer from poor sensitivity compared to conventional DNA analysis in a forensic laboratory. In order to save time, work-up steps such as extraction and/or purification are skipped. When taking all the mentioned parameters into account, the ANDE system performs slightly better than the RapidHIT ID system. Although the ANDE system is somewhat heavier, it has a better sensitivity range. It can run more samples at the same time. The analysis time of both systems is comparable. 

Overall, impressive developments have taken place. Nevertheless, there is still no portable device containing all the required steps available for fast analysis (under 1 h for a full STR profile) of real ‘dirty’ samples directly at the crime scene. It would also be ideal to have a multi-compartment chip, which makes it possible to perform fast and reliable analyses directly at the crime scene, as well as on-chip storage of samples for (further) laboratory analyses.

## Figures and Tables

**Figure 1 sensors-23-01075-f001:**
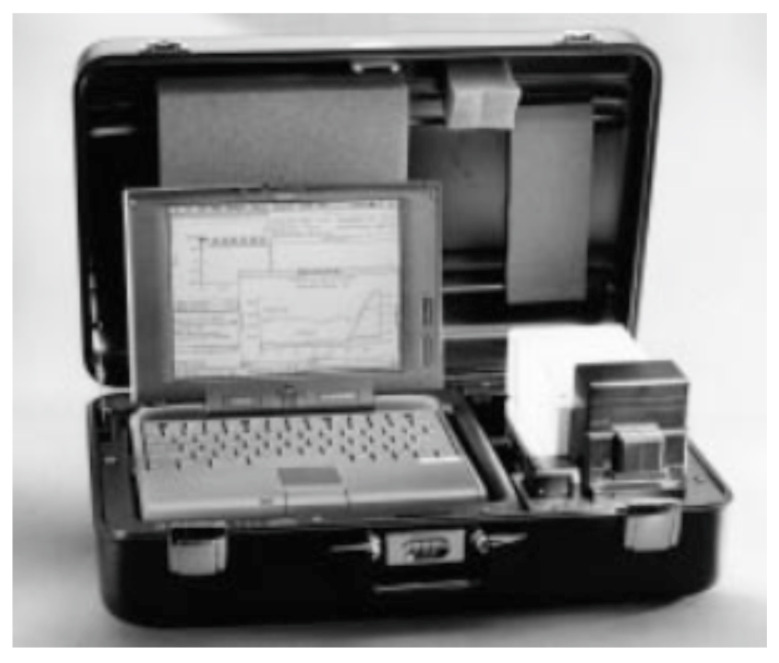
A photograph of the MATCI, which was the size of a briefcase and was rechargeable battery-operated. Reprinted (adapted) with permission from [13]. Copyright 1998 American Chemical Society.

**Figure 2 sensors-23-01075-f002:**
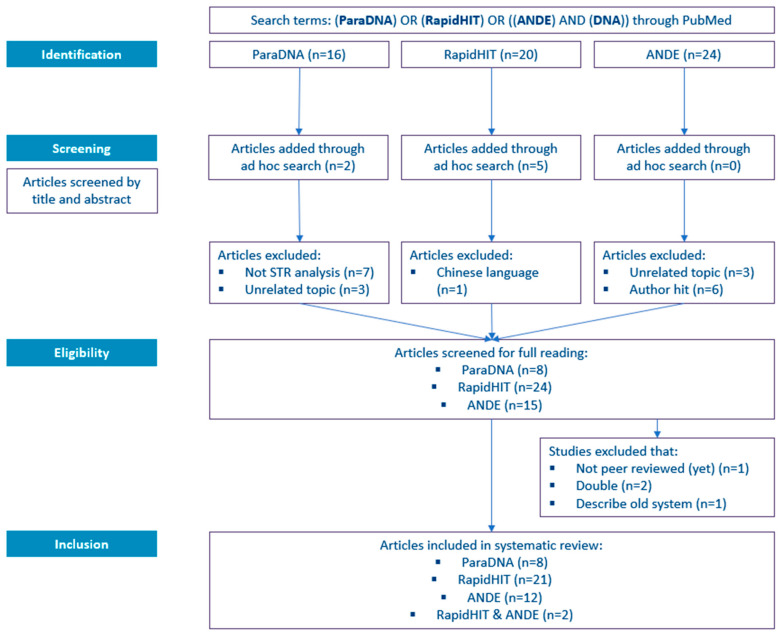
The PRISMA flow diagram was used in this systematic review.

**Figure 3 sensors-23-01075-f003:**
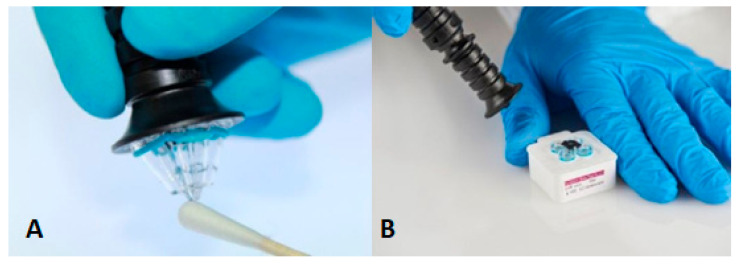
(**A**) The sample collector of the ParaDNA system, (**B**) The nibs of the sample collector inserted into the reaction plate [41]. Reprinted from Forensic Science International: Genetics Supplement Series, 6, Copyright (2017), with permission from Elsevier.

**Figure 4 sensors-23-01075-f004:**
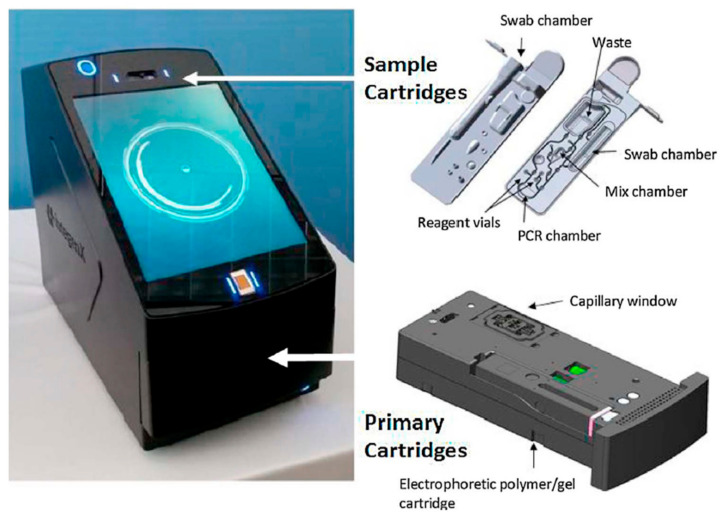
The RapidHIT ID system from IntegenX with a primary cartridge (a reagent cartridge that contains the necessary reagents and consumables for capillary electrophoresis) and a sample cartridge [4]. Reprinted from Forensic Science International: Genetics, Copyright (2017), with permission from Elsevier.

**Figure 5 sensors-23-01075-f005:**
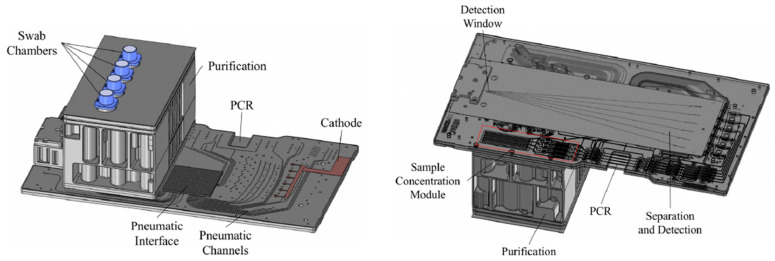
Schematic overview of the ANDE I-chip, with a top view (**left**) and a bottom view (**right**). Reprinted from [81].

**Figure 6 sensors-23-01075-f006:**
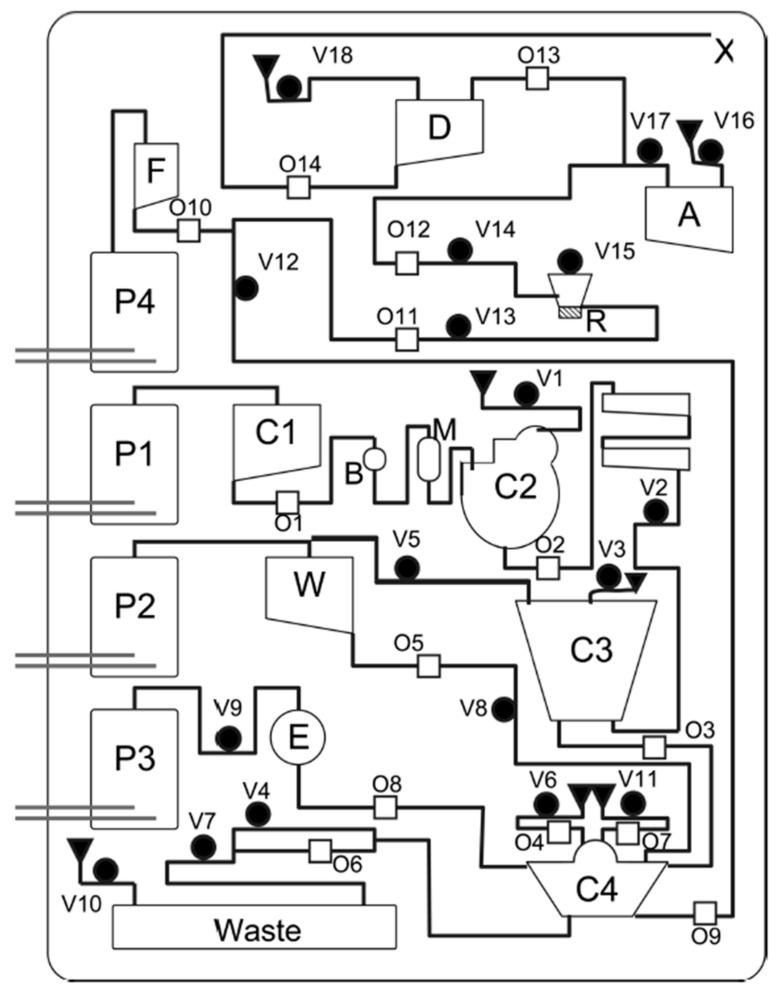
Overview of the MiDAS cartridge for DNA extraction, amplification, and post PCR denaturation, with (P1–P4) electrochemical pumps; (C1) lysate input chamber; (C2) bead chamber; (C3) mixing/incubation chamber; (C4) washing and elution chamber; (R) PCR chamber; (A) DNA extract archive chamber; (D) denaturation chamber; (M) bead storage chamber; (B) binding buffer chamber; (W) wash solution storage chamber; (E) elution buffer storage chamber; (F) Formamide/ILS storage chamber; (X) output to CE. A closing valve is represented by b; an opening valve by 0; and a vent by 1. Reprinted (adapted) with permission from [14]. Copyright 2010 American Chemical Society.

**Table 1 sensors-23-01075-t001:** Characteristics of the ParaDNA, RapidHIT 200, RapidHIT ID and ANDE systems.

	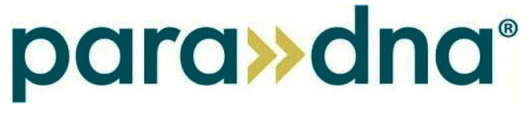	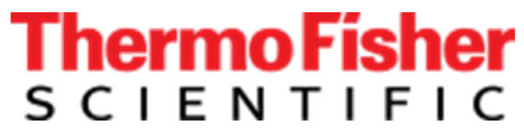		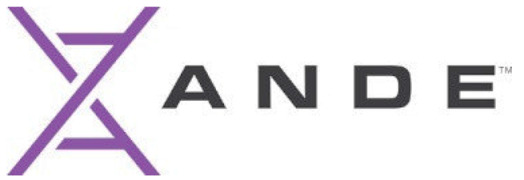
	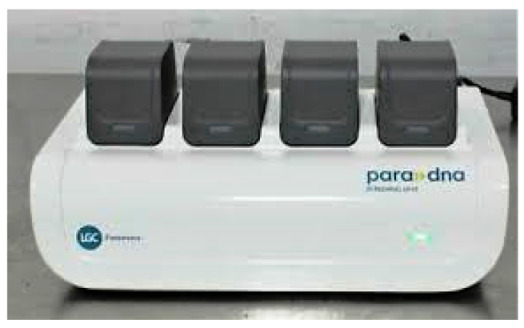	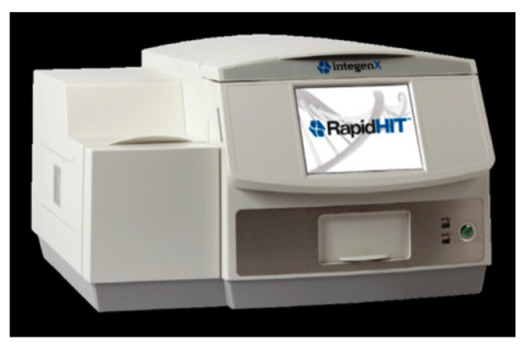	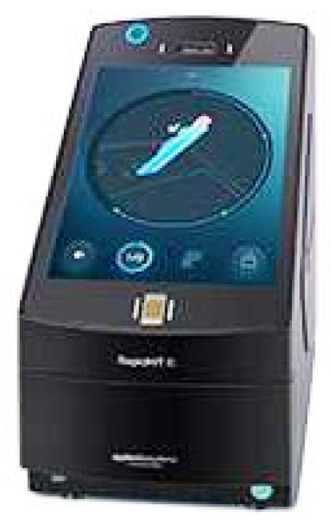	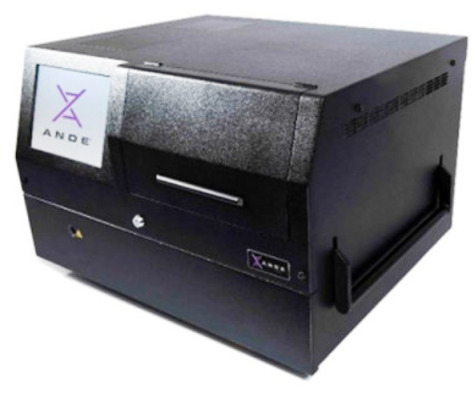
	**ParaDNA**	**RapidHIT 200**	**RapidHIT ID**	**ANDE 6C**
**Manufacturer**	LGC Forensics	IntegenX (Thermo Fisher Scientific)		ANDE
**Weight**	6 kg	81.5 kg	29 kg	54 kg
**Dimensions** **(L× W × H)**	39 × 29 × 19 cm^3^	73 × 71 × 48 cm^3^	135 × 122 × 69 cm^3^	75 × 45 × 60 cm^3^
**# STRs**	6 loci	GlobalFiler Express: 24 loci AmpFLSTR NGM SElect Express: 17 loci		A-chip: 27 loci I-chip: 27 loci
**Sampling**	ParaDNA sample collector	GlobalFiler Express: Swab (Puritan Cotton Swab) AmpFLSTR NGM SElect Express: Swab (Whatman OmniSwab)		A-Chip: (Buccal) Swab I-Chip: Swab
**Technology**	HyBeacons	CE		CE
**Time-to-result**	75 min	<2 h	90 (−110) min	A-chip: 94 min I-chip: 106 min
**# Samples**	4	Up to 5 *	1	A-chip: 5 I-chip: 4
**Sensitivity**	Not enough data	50–200 ng	40–80 ng	250 ng-2 µg
**Notes**	Discontinued	Replaced by the RapidHIT ID		
	Only indicative			

* The cartridge of the RapidHIT has eight positions, including the positive and negative control and the allelic ladder [56].

## Data Availability

Not applicable.

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
