# Peer review of "A Systematic Review on Commercially Available Integrated Systems for Forensic DNA Analysis"

_sensors, 2023, doi:10.3390/s23031075_

Round 1

Reviewer 1 Report

The review compares commercially available integrated systems for forensic DNA analysis. The review is reasonably comprehensive, with most of the focus placed on the performance of the systems regarding times, costs, reliability etc. There is less focus placed on the technologies themselves and how this effects the performance. For example, it would be interesting to hear more about where the bottlenecks are for performance in the different methodologies used in the systems and where they could be improved.

I found the review informative and interesting. However, I struggled with the language. The manuscript requires a considerable rewrite by an English expert, or native speaker. 

When the authors state that something is recommended, (page 10 paragraph 5) they should clearly state by who.

It is also less interesting to hear that “not a wide variety of research groups reported…”. It would be better to give a clear description of who did report and comment on whether or not this is a representative group. This appears in several places in the manuscript.

The authors should avoid statements like, “This is a time consuming process, which takes precious time.” Be more specific. How long does it take? what is the impact of this delay?

With so many product names and different models, the authors should really try and avoid confusion by being as clear as possible about when kits and models are the same or different. For example, the authors mention; AmpFISTR NGM SElect, AmpFISTR NGMSeE and AmpFISTR1 NGMSElect Express on pages 8 and 9. Are these the same? Are they typos?  

Author Response

See attachment for our point-by-point response to reviewer #1.

Reviewer 2 Report

I would like to thank the authors for the manuscript they produces. They did and excellend job in reviewing the available integrated systems for forensic DNA analysis.

Author Response

See attachment for our point-by-point response to reviewer #2.

Reviewer 3 Report

In general, I consider it an interesting study. Nevertheless, I have made some observations that must be taken into account, all my comments are collected in the attached file.

Author Response

See attachment for our point-by-point response to reviewer #3.

Round 2

Reviewer 1 Report

Dear authors, thank you for considering my comments and for your carefully composed replies. I would be happy to see the manuscript accepted, although I stand by my opinion that it would be greatly improved by a quick run through by a native English speaker. I do not have the time to do this. It is not within my remit to edit language, but to point out that it could be improved. It is understandable as is, but could be much better with a relatively small effort. 

However, as the authors asked for some examples, here are a few from the first page:

Page 1, Line 25: “Sampling and securing traces at a crime scene is a crucial step in the investigation process, in The Netherlands generally carried out by the police.” There are two steps mentioned and so the plural ("are" rather than "is" should be used). “Sampling and securing traces at a crime scene are crucial steps in the investigation process, in The Netherlands generally carried out by the police.”

Page 1, line 36: “Technical innovations like the miniaturization of conventional laboratory techniques can improve the analysis speed as well as might enable use directly at the crime scene.” This just sounds wrong to a native English speaker. Better as “Technical innovations like the miniaturization of conventional laboratory techniques may both improve their speed and enable their use directly at the crime scene.”

Page 1, line 45: “…Next step is the sample work-up whereby the cells are eluted and lysed and subsequently the DNA will be extracted…” this is a mixing of tenses (is, and will be). Would be better as “…The following step is the sample work-up whereby cells are eluted and lysed, and DNA is extracted…”

Author Response

See attachment for our response.
